# SDSC: A Structure-Aware Metric for Semantic Signal Representation Learning

## Abstract

We propose the Signal Dice Similarity Coefficient (SDSC), a structure-aware reconstruction metric for time-series self-supervised representation learning. Here, structure-aware refers to the local waveform consistency characterized by sign and magnitude overlap, rather than global temporal alignment. SDSC replaces the MSE loss only in the reconstruction branch of SimMTM, while its contrastive objective remains unchanged. This design enables a fair analysis of how structure-aware reconstruction affects representation quality without altering the contrastive learning process. Most existing methods rely on distance-based objectives such as mean squared error, which are sensitive to amplitude, often under-penalize waveform polarity, and unbounded in scale. These properties hinder semantic alignment and reduce interpretability. The SDSC quantifies structural agreement between temporal signals based on signed amplitude intersections, extending the Dice Similarity Coefficient from segmentation to continuous signals. Although originally defined as a metric, SDSC can also be used as a loss by subtracting from 1 and applying a differentiable Heaviside approximation. A hybrid loss that combines SDSC with MSE further improves stability and preserves amplitude when necessary. Experiments on forecasting and classification benchmarks demonstrate that SDSC-based pre-training achieves comparable or improved performance relative to MSE, particularly in in-domain and low-resource scenarios. These results suggest that enforcing structural fidelity enhances semantic representation quality and motivates the reconsideration of structure-aware objectives as alternatives to conventional distance-based losses.

## 1 Introduction

Self-Supervised Learning(SSL) enables representation learning from unlabeled data by formulating proxy objectives such as masked prediction or reconstruction. In Computer Vision (CV) and Natural Language Processing (NLP), SSL methods have demonstrated effective feature extraction capabilities. (Gui et al., 2024)

In time-series modeling, SSL is also applied to extract semantic representations for downstream tasks, including forecasting and classification. (Ma et al., 2024) In time-series signals such as EEG and EMG, task-relevant semantics are often encoded in structural features, including waveform shapes, phase alignment, and local frequency patterns. Signal-specific representations require attention to these temporal structures. However, most of the reconstruction-based SSL methods commonly adopt distance-based metrics such as mean squared error(MSE), which focus on amplitude differences and element-wise deviations. Distance-based metrics exhibit sensitivity to scale and are invariant to waveform polarity, resulting in reconstructions that minimize error without preserving structural consistency. In addition, distance-based metrics often assign low values to reconstructions that differ semantically from the target. Phase-inverted signals, amplitude-scaled signals, and zero-valued baselines produce similar MSE values despite large semantic deviations. Moreover, metric values are unbounded and non-normalized, which complicates interpretation and model selection. Therefore, the lack of interpretability and amplitude bias in MSE limit its reliability. Recent studies suggest that the default reliance on MSE may not be optimal for all time-series tasks (Zeng et al., 2023).

To address these issues, a novel structure-aware metric, the Signal Dice Similarity Coefficient (SDSC), is introduced. The SDSC is inspired by the Dice Similarity Coefficient(DSC) (Dice, 1945; Sørensen, 1948) widely used in semantic segmentation tasks and quantifies the structural agreement between two temporal signals. The metric is bounded in [0, 1], reduces sensitivity to amplitude variation, and reflects alignment in waveform structure. In addition, the metric is implemented as a loss by subtracting values from 1 and a differentiable approximation of the Heaviside function is applied to enable gradient-based optimization. Lastly, a hybrid loss that combines SDSC and MSE is introduced to enhance stability during training. Experiments on forecasting and classification benchmarks show that SDSC-based pre-training achieves comparable or improved performance relative to MSE, particularly when encoders are fixed. Performance gains are observed in in-domain and low-resource settings, where structural fidelity in the representations contributes to downstream accuracy.

Self-supervised learning for time-series signals typically consists of two components: reconstruction-based and contrastive learning. While contrastive objectives explicitly enforce instance discrimination, reconstruction-based objectives aim to recover the signal content. In this context, the choice of reconstruction loss directly influences the quality of the learned representation. Despite their widespread use, distance-based losses do not reflect the semantic structure and amplify the sensitivity to signal amplitude. The empirical similarity in downstream performance between MSE and SDSC indicates that MSE-based models achieve competitive results not due to accurate semantic preservation but due to incidental alignment with signal structure. In this study, SDSC is integrated exclusively into the reconstruction branch of SimMTM, a self-supervised framework composed of both contrastive and reconstruction objectives. The contrastive loss (InfoNCE) (Oord et al., 2018) is kept identical to the original SimMTM formulation. The term structure-aware in this paper specifically denotes local structural similarity captured by pointwise sign agreement and magnitude overlap. SDSC is therefore alignment-free and computationally linear, but not tolerant to global shifts or warping. This setup isolates the contribution of the reconstruction loss and allows a clean comparison under identical contrastive conditions.These observations indicate that distance-based metrics might not fully capture the semantic structure of signals, motivating consideration of a shift towards structure-aware representation learning paradigms.

## 2 RELATED WORKS

Recent work has questioned the effectiveness of transformers for time-series forecasting (Zeng et al., 2023). This paper is related to the Dice Score Coefficient (DSC) (Dice, 1945; Sørensen, 1948), and the broader field of time-series modeling (TSM), specifically time-series pre-training models (TS-PTM).

### 2.1 EVALUATION METRICS

Traditional TS-PTM research widely used reconstruction metrics such as MSE or MAE. These metrics ignore temporal misalignments, limiting their effectiveness. DTW (Sankoff & Kruskal, 1983) addressed misalignment but is computationally expensive. FastDTW (Salvador & Chan, 2007) reduced complexity but is not differentiable and thus unsuitable for training. SoftDTW (Cuturi & Blondel, 2017) provided a differentiable approximation, making it usable in training. All remain distance-based. DILATE (Le Guen & Thome, 2019) combines shape and temporal distortion losses, but is limited to forecasting.

Other alternatives target correlation. The Pearson Correlation Coefficient (PCC) (Bishop & Nasrabadi, 2006) measures linear dependence but is sensitive to phase shifts and mainly reflects point-wise similarity. In audio, Scale-Invariant SNR (SI-SNR) (Luo & Mesgarani, 2018) is used as a structure-aware objective, but it only maximizes signal-to-error ratio rather than comparing shapes directly.

The Dice Score Coefficient (DSC) (Dice, 1945; Sørensen, 1948) is widely used in semantic segmentation for overlap-based evaluation. We extend this idea to time-series and propose the SDSC. The SDSC directly measures structural overlap, emphasizing waveform shape rather than amplitude, and can serve as both a metric and a training loss. A summary of alignment-based objectives and our SDSC in terms of complexity and properties is provided in the Appendix A.1.

## 2.2 TIME-SERIES MODELING

According to (Gui et al., 2024), SSL typically follows a two-stage process: unsupervised pre-training followed by task-specific fine-tuning. This paradigm has driven major advances in time-series learning (Ma et al., 2024).

Several TS-PTM methods focus on representation learning. TS2Vec (Yue et al., 2022) learns contextual representations, while CoST (Woo et al., 2022) applies contrastive objectives. TimesNet (Wu et al., 2022) targets general time-series analysis, and (Eldele et al., 2023) study semi-supervised settings. TI-MAE (Li et al., 2023) introduces masked autoencoders, and SimMTM (Dong et al., 2023) simplifies the masked framework. iTransformer (Liu et al., 2023) uses inverted transformers for efficient forecasting, while the Unified Transformer (Woo et al., 2024) supports multiple downstream tasks. TIMER (Liu et al., 2024) develops generative pre-trained transformers. TimeSiam (Dong et al., 2024) employs Siamese networks, and TimeDiT (Cao et al., 2024) explores diffusion-based pre-training. Cross-domain modeling has also been studied (Prabhakar Kamarthi & Prakash, 2024), and (He et al., 2025) propose universal representations.

Most of these advances rely on architectural design or contrastive strategies. Such approaches improve performance but are limited in capturing structural similarity. Unlike distance-based or alignment-based metrics, SDSC directly and efficiently quantifies structural similarity, addressing a crucial gap in representation learning for time-series. SimMTM (Dong et al., 2023) is chosen as the baseline because it combines contrastive and reconstruction objectives in a modular manner. In our setup, only the reconstruction loss (MSE) is replaced by SDSC, while the contrastive component (InfoNCE) (Oord et al., 2018) remains fixed. This controlled configuration ensures that observed performance differences originate from the reconstruction objective itself rather than from contrastive learning effects.

## 3 SIGNAL DICE SIMILARITY COEFFICIENT

In this section, we introduce the SDSC, a new metric designed to explicitly quantify structural similarity between two signals. In signal representation learning, reconstructing signals accurately is important to capture their meaning.

Table 1: Quantitative evaluation of signal variations using distance-based metrics and the proposed SDSC.

| Signals | MSE↓ | MAE↓ | DTW↓ | SDSC↑ |
|---|---|---|---|---|
| Inverted | 0.0200 | 0.1272 | 0.0425 | 0.0000 |
| 0.5x Scaled | 0.1249 | 0.3180 | 0.1353 | 0.6667 |
| 2x Scaled | 0.4995 | 0.6360 | 0.2706 | 0.6667 |
| Zero | 0.4995 | 0.6360 | 0.6360 | 0.0000 |
| Noise Sample | 0.5062 | 0.6361 | 0.2236 | 0.1137 |
| Positive Shifted | 1.0000 | 1.0000 | 0.6228 | 0.3887 |
| Negative Shifted | 1.0000 | 1.0000 | 0.6228 | 0.3887 |

## 3.1 DISTANCE-BASED METRICS

Distance-based metrics such as MSE, mean absolute error (MAE) and dynamic time warping (DTW) are widely used to measure the difference between predicted and ground-truth signals. These metrics evaluate element-wise deviations and are effective in reducing numerical reconstruction errors. However, distance-based metrics focus primarily on signal amplitude and do not consider the polarity or structural shape of the waveform.

Figure 1 and Table 1 illustrate examples that expose key limitations of distance-based metrics. Figure 1a illustrates a complete phase inversion under low-amplitude conditions, visually preserving the waveform shape while reversing its polarity. As shown in Table 1 (Inverted), the inverted signal receives low error scores on all distance-based metrics (for example, MSE = 0.0200), making it

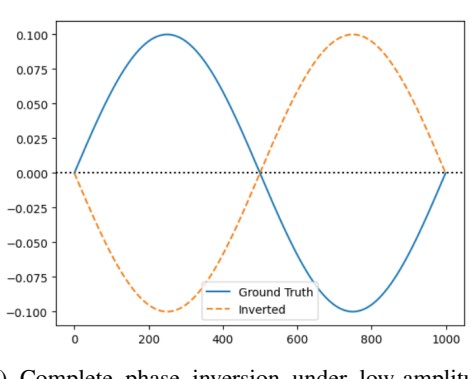 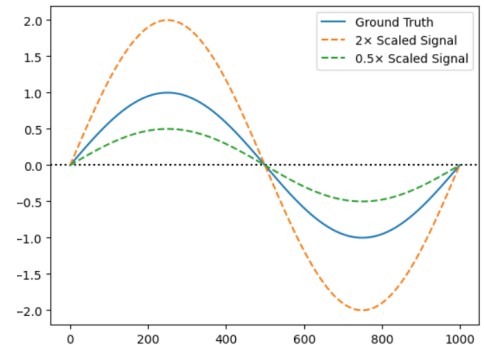

(a) Complete phase inversion under low-amplitude conditions.

(b) 0.5× and 2× scaled signals introducing structural distortions.

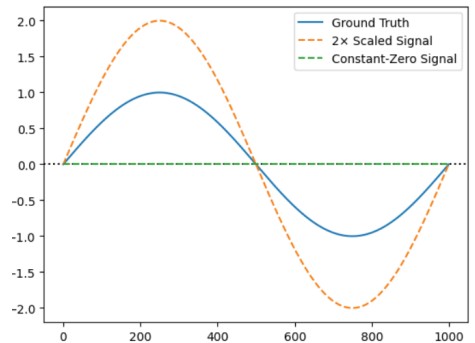 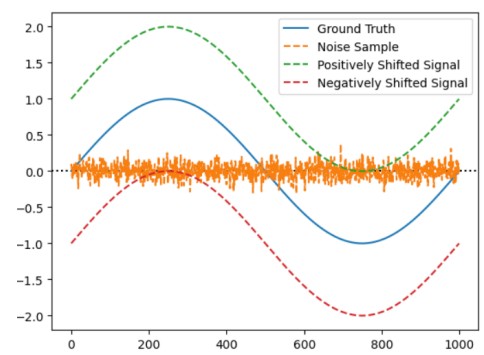

(c) Constant zero signal vs. 2× scaled waveform with identical MSE.

(d) Noisy outputs with semantically valid yet shifted waveforms.

Figure 1: Examples demonstrating the limitations of distance-based metrics in capturing structural similarity. SDSC offers a more faithful assessment in (a) phase-shifted signals, (b) scale-induced distortions, (c) structurally dissimilar but MSE-equivalent signals, and (d) noisy outputs with underestimated errors.

appear as a high quality reconstruction despite its semantic reversal. Figure 1b compares 0.5× and 2× scaled signals, both of which introduce comparable structural distortions but produce markedly different metric values due to amplitude differences. As indicated in Table 1, the dependence on the amplitude obscures the true degree of structural deviation, resulting in an inaccurate evaluation of the signal quality. In Figure 1c, a constant zero signal is evaluated alongside a 2× scaled waveform. As shown in Table 1, both produce identical MSE scores (0.4995), despite their stark structural differences. The similarity in MSE scores despite structural differences reveals the inability of MSE to distinguish between waveforms when average magnitudes are equivalent. Lastly, Figure 1d contrasts noisy outputs with semantically valid yet shifted waveforms. The noise-dominated signal produces an MSE (0.5062) comparable to semantically valid signals due to its fluctuation averaging around the baseline, making it appear deceptively accurate under distance-based metrics. Although numerically favorable, the output is structurally misaligned and functionally misleading. Such insensitivity to signal semantics is particularly problematic for physiological data like EEG or ECG, where subtle structural components often carry diagnostic significance. Therefore, exclusive reliance on amplitude-centric metrics may lead to semantically incorrect reconstructions.

## 3.2 DEFINITION OF SDSC

The SDSC extends the DSC, commonly used for set overlap in semantic segmentation, to continuous, signed time-series data. This extension is motivated by the observation that the goal of signal reconstruction in SSL is not merely to minimize the amplitude error, but to restore the signal's underlying shape, a concept that is difficult to formalize directly. We propose using the area under the curve as a tractable proxy for waveform shape. This reframes the problem of comparing two

signal shapes as measuring the overlap between their respective areas. This area overlap problem is analogous to the well-posed problem of measuring pixel overlap in semantic segmentation, making the Dice Similarity Coefficient (DSC) a natural and theoretically sound foundation for our metric. Instead of relying on the membership of the sets, SDSC computes structural alignment from signed amplitude intersections at each time step. This formulation captures polarity agreement and local magnitude overlap, which represent the local structural consistency of the signals. It implicitly reflects small phase variations but does not account for temporal shifts or warping. Like DSC, SDSC returns a score in the range $[0, 1]$. The original DSC measures set the similarity as follows:

$$DSC = \frac{2|X \cap Y|}{|X| + |Y|} \tag{1}$$

Here, $|X|$ and $|Y|$ denote the cardinalities of the respective sets, and the metric reflects the size of their intersection relative to the total area. The SDSC extends this concept to the signal domain by interpreting the area under the curve as a proxy for the waveform structure. Given two signal functions $E(t)$ and $R(t)$ representing ground truth and reconstruction, the SDSC is defined as :

$$S(t) = E(t) \cdot R(t) \tag{2}$$

$$M(t) = min(|E(t)|, |R(t)|) = \frac{|E(t)| + |R(t)| - ||E(t)| - |R(t)||}{2} \tag{3}$$

$$SDSC(E(t), R(t)) = \frac{2 \cdot \int H(S(t)) \cdot M(t)\, dt}{\int (|E(t)| + |R(t)|)\, dt} \tag{4}$$

$H(\cdot)$ denotes the Heaviside step function(see in Appendix A.2), and $t \in T$ is given time. The objective in signal representation learning is to maximize Equation (4) toward 1. However, directly computing SDSC via integration is infeasible in practice, as real-world signals, such as EEG, lack known analytical expressions. To address this, a discrete approximation is adopted.

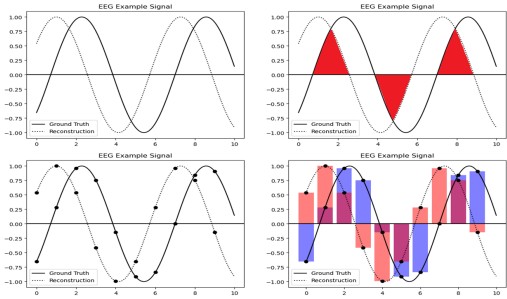

Figure 2: Example of intersection between two signals and discrete approximation.

Figure 2 illustrates the approximation procedure. Although signals are continuous in nature, real-world signals are typically sampled at uniform intervals. Consequently, each sampled value is treated as a rectangle of unit width, allowing the continuous integral to be approximated by summation.

$$SDSC(E(t), R(t)) \approx \frac{2 \cdot \sum (H(S(s)) \cdot M(s))}{\sum (|E(s)| + |R(s)|) + \epsilon} \tag{5}$$

where $s \in S$ are discrete sampling points, with $S \subset T$ and $\epsilon$ is a small constant to prevent division by zero. The proposed approximation enables a tractable computation of SDSC on real signals and ensures a consistent evaluation. As demonstrated in Table 1, the SDSC shows increased robustness to polarity shifts and amplitude scaling. As proven in Lemma 1, the proposed SDSC is bounded in the range [0, 1] and facilitates standardized interpretation in all signal domains. Unlike MSE or DTW, the SDSC is less affected by signal magnitude, thereby reducing distortions due to scale and polarity. The normalized form also simplifies cross-domain comparisons, enabling a more structure-aware assessment of signal reconstruction quality.

### 3.3 HYBRID LOSS INTEGRATION

Since the SDSC score is bounded in $[0, 1]$, we can define the loss as $1 - SDSC(\cdot)$.

$$\mathcal{L}_{sdsc} = 1 - SDSC(E(S), R(S)) \tag{6}$$

However, the use of the Heaviside step function in Equation (4) introduces discontinuities, which can negatively affect the stability of training. Continuity is preserved when at least one of the signals maintains the same sign at the corresponding sampled points. However, the likelihood of sign mismatches increases when the sampling resolution is low. To enable stable gradient-based optimization, a smooth approximation of the Heaviside function is introduced. The following sigmoid-based formulation is used, with a sharpness parameter $\alpha$.

$$\hat{H}(x) = \frac{1}{1 + e^{-\alpha x}} \tag{7}$$

If the sharpness parameter $\alpha$ is large, the sigmoid-based approximation $\hat{H}(x)$ more closely resembles the original Heaviside function. However, excessively large values of $\alpha$ can lead to sharp transitions that result in unstable gradients, potentially damaging the training process.

SDSC captures structure but ignores amplitude, whereas MSE captures amplitude but misses structure. To balance the strengths of both approaches, this work proposes a hybrid loss function that combines the structural awareness of SDSC with the amplitude sensitivity of MSE. The final objective function is formulated as follows:

$$\mathcal{L}_{hybrid} = \lambda_{sdsc} \cdot \mathcal{L}_{sdsc} + \lambda_{mse} \cdot \mathcal{L}_{MSE} \tag{8}$$

Here, $\lambda_{sdsc}$ and $\lambda_{mse}$ are parameters that control the trade-off between structural accuracy and amplitude-based accuracy. To determine these weights, we adopt the uncertainty-based tuning strategy proposed in (Kendall et al., 2018), where the weighting coefficients are adapted based on the homoscedastic uncertainty associated with each loss term. This hybrid formulation promotes reconstructions that are structurally aligned and numerically precise. In practice, each loss term's weight is parameterized by a trainable log-variance term following Kendall et al. (2018), and updated jointly with model parameters to balance the relative homoscedastic uncertainty of $\mathcal{L}_{sdsc}$ and $\mathcal{L}_{mse}$.

The overall pre-training objective of SimMTM consists of a contrastive term $\mathcal{L}_{con}$ (InfoNCE) (Oord et al., 2018) and a reconstruction term $\mathcal{L}_{rec}$. We keep $\mathcal{L}_{con}$ identical to the original SimMTM formulation and replace $\mathcal{L}_{rec}$ with either MSE, SDSC, or the proposed Hybrid loss. The total loss is given by:

$$\mathcal{L}_{total} = \mathcal{L}_{con} + \mathcal{L}_{rec}, \tag{9}$$

where $\mathcal{L}_{rec} \in \{\mathcal{L}_{MSE}, \mathcal{L}_{sdsc}, \mathcal{L}_{hybrid}, \mathcal{L}_{pcc}, \mathcal{L}_{si\_snr}, \mathcal{L}_{softdtw}\}$. This formulation isolates the effect of the reconstruction objective under a fixed contrastive setup.

## 4 EXPERIMENTS

All experiments are conducted with fixed random seeds across all runs to ensure reproducibility. The contrastive objective of SimMTM is kept unchanged in all experiments. Therefore, any downstream performance differences should be attributed to the reconstruction objective (MSE vs. SDSC/Hybrid), not to modifications in the contrastive component. Training and evaluation are performed on NVIDIA 3090 GPUs (2×). For a controlled comparison, we adopt SimMTM (Dong et al., 2023) as the sole backbone model, as its architectural simplicity allows for a clear analysis of the impact of different loss functions. Although conceptually lightweight, SimMTM internally employs transformer-based encoders with multi-head self-attention and temporal masking, similar to recent pretraining models such as PatchTST (Nie, 2022). In fact, SimMTM was reported in the NeurIPS 2023 benchmark suite to outperform several transformer backbones, including PatchTST (Nie, 2022), demonstrating that its simplicity lies in framework design rather than model capacity. Therefore, using SimMTM provides a fair and expressive foundation to isolate the effect of the reconstruction objective without confounding factors from architectural differences. SDSC replaces the MSE

reconstruction loss, but the contrastive objective of SimMTM is unchanged. This setup isolates the reconstruction loss contribution from contrastive effects, providing a fair comparison. In our pre-training experiments, we compare our proposed SDSC not only with the conventional MSE but also with other structure-aware objectives like PCC (Bishop & Nasrabadi, 2006) and SI-SNR (Luo & Mesgarani, 2018). Baseline models are reproduced using their official implementations, The detailed hyperparameter settings for all experiments, including our choice of $\alpha = 10$ for SDSC(based on the analysis in Appendix A.3), are provided in Appendix A.4. For time-series forecasting tasks, we adopt MSE and MAE as evaluation metrics. For classification experiments, accuracy, precision, recall, and F1 score are computed, along with their macro-average values. All time-series inputs are z-score normalized per channel using statistics computed only from the training split, ensuring both scale consistency and the removal of DC offsets without any leakage. We also report a controlled evaluation using frozen $\lambda = 0.5$ to verify that the observed trends are not driven by hyperparameter tuning. The corresponding results are provided in Appendix A.6, A.8, A.10, A.13.

## 4.1 PRE-TRAINING

Table 2: Summary of pre-training performance averaged across forecasting and classification datasets. SDSC shows the most robust results overall (full results in Appendix A.5, A.7). Note: SI-SNR values use a different scale and sometimes fail to converge (e.g., ETTh1), so they are reported for completeness.

| Dataset | Loss | MSE↓ | MAE↓ | SDSC↑ |
|---|---|---|---|---|
| **Avg (Forecasting)** | MSE | 0.4852 | 0.3525 | 0.7670 |
| | SoftDTW | 1.3273 | 0.7432 | 0.4990 |
| | PCC | 1.3289 | 0.6705 | 0.5274 |
| | SI-SNR | 34.9085 | 2.5408 | 0.4523 |
| | **SDSC(Ours)** | 0.6348 | 0.3870 | 0.7723 |
| | **Hybrid(Ours)** | **0.4783** | **0.3368** | **0.7841** |
| **Avg (Classification)** | MSE | 50.3203 | 3.5269 | 0.6105 |
| | Soft-DTW | **49.1339** | **3.4751** | 0.6210 |
| | PCC | 120.0105 | 4.5091 | 0.1622 |
| | SI-SNR | 118.6110 | 4.4846 | 0.1693 |
| | **SDSC(Ours)** | 74.0253 | 3.8626 | **0.6610** |
| | **Hybrid(Ours)** | 50.3471 | 3.5286 | 0.6481 |

In the pre-training phase, we compare three objective functions for SSL: MSE loss, SDSC loss, and a hybrid loss that combines both. Table 2 summarizes the results in both the forecasting and classification datasets.

MSE-based models achieve lower reconstruction errors under distance-based metrics, as expected. In contrast, SDSC-based models achieve higher SDSC scores, which are indicative of structural alignment. While the metrics are weakly correlated, each captures distinct aspects of the signal, motivating a closer examination of their downstream impact.

Figure 3a shows the relationship between MSE and SDSC for models pre-trained on ETTh1 using MSE. SDSC increases as MSE decreases, but the Pearson correlation is $-0.324$, indicating a weak alignment. Figure 3b, Figure 3c compare the SDSC distributions at fixed MSE ($1.5 \pm \epsilon$) under MSE-based and SDSC-based pre-training. SDSC-based pre-training achieves higher SDSC values at the same MSE level.

Table 3: Comparison of SDSC concentration under a fixed MSE ($1.5 \pm \epsilon$).

| Model Type | Std Dev | IQR |
|---|---|---|
| MSE-based model | 0.0280 | 0.0418 |
| SDSC-based model | 0.0249 | 0.0384 |

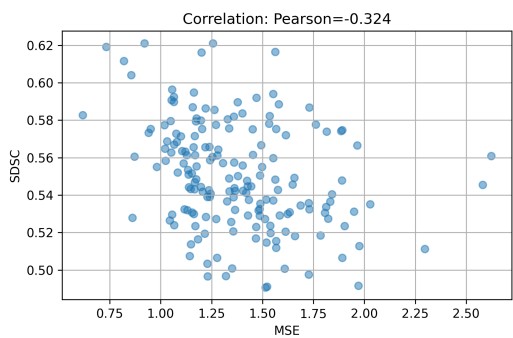

(a) Scatter plot between MSE and SDSC under MSE-based pre-training.

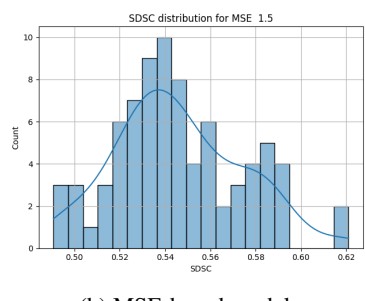

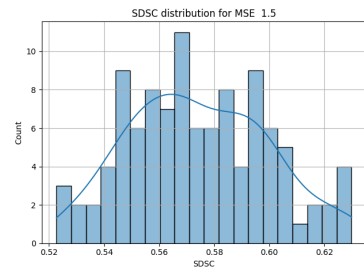

(b) MSE-based model.

(c) SDSC-based model.

Figure 3: Analysis of structural alignment in MSE-based pre-training. (a) The weak negative correlation between MSE and SDSC suggests limited alignment. (b, c) At a fixed MSE of $1.5 \pm \epsilon$, the SDSC-based model achieves a distribution with higher structural scores.

Table 3 reports the standard deviation and interquartile range (IQR) of SDSC values under fixed MSE. SDSC-based models exhibit lower variance and tighter concentration, suggesting a more consistent structural alignment.

These results indicate that MSE-based SSL captures structural features to some extent but lacks reliability. The weak correlation explains why SDSC values remain low when models are trained solely with MSE. SDSC-based training enhances structural fidelity but reduces amplitude precision, resulting in higher MSE. The hybrid loss addresses this trade-off and achieves stable performance across both metrics.

## 4.2 FORECASTING

The forecasting task is performed in an in-domain setting using models pre-trained with MSE, SDSC, and hybrid objectives. For consistency, the evaluation is conducted at the best epoch for both pre-training and fine-tuning. We adopt MSE and MAE as evaluation metrics, following standard practice in forecasting tasks.

Table 4 summarizes the results of the forecasting task. As shown in Appendix A.5, pre-trained MSE models learn SDSC to some extent, and pre-trained SDSC models, although the SDSC model has higher reconstruction errors, achieve similar downstream accuracy. These results indicate that beyond a certain threshold, additional MSE reduction achieves diminishing returns, suggesting limited benefits of focusing solely on amplitude rather than structural alignment. SDSC models achieve a similar accuracy with significantly higher MSE, implying that structural alignment alone suffices for downstream prediction.

Table 4: Forecasting fine-tuning performance on a representative dataset (Electricity) and the average across all datasets.(Full results in Appendix A.9.)

| Dataset | Pre-training Loss | MSE↓ | MAE↓ |
|---|---|---|---|
| Electricity | MSE | 0.200 | 0.291 |
| | Soft-DTW | 0.200 | 0.290 |
| | PCC | 0.199 | **0.288** |
| | SI-SNR | 0.203 | 0.292 |
| | **SDSC(Ours)** | 0.200 | 0.293 |
| | **Hybrid(Ours)** | **0.198** | **0.288** |
| Avg | MSE | 0.295 | **0.316** |
| | Soft-DTW | 0.303 | 0.322 |
| | PCC | 0.296 | 0.318 |
| | SI-SNR | 0.310 | 0.325 |
| | **SDSC(Ours)** | **0.294** | **0.316** |
| | **Hybrid(Ours)** | **0.294** | **0.316** |

## 4.3 CLASSIFICATION

The classification task is performed on both in-domain and cross-domain classification tasks using pre-trained encoders. To assess the effect of pre-training, we compare models with frozen encoders and those with end-to-end fine-tuning. All models are evaluated using accuracy, precision, recall, and the F1 score.

Table 5: Summary of freeze classification performance, averaged across scenarios. Our proposed SDSC shows notable improvements in the in-domain setting. (full results in Appendix A.11)

| Scenario | Loss | Acc.↑ | Prec.↑ | Rec.↑ | F1↑ | Avg↑ |
|---|---|---|---|---|---|---|
| **Avg (In Domain)** | MSE | 75.45 | 73.07 | 63.49 | 64.59 | 69.15 |
| | Soft-DTW | 68.76 | 40.89 | 47.36 | 43.52 | 50.13 |
| | PCC | 71.40 | 47.38 | 51.56 | 46.70 | 54.26 |
| | SI-SNR | 71.08 | 47.40 | 51.81 | 46.91 | 54.30 |
| | **SDSC(Ours)** | **76.38** | **74.19** | **64.93** | **65.85** | **70.34** |
| | **Hybrid(Ours)** | 76.23 | 72.72 | 65.61 | 66.47 | 70.26 |
| **Avg (Cross Domain)** | MSE | **62.19** | 39.04 | **47.89** | 41.42 | 47.63 |
| | Soft-DTW | 62.00 | **41.47** | 47.75 | **41.73** | **48.24** |
| | PCC | 60.47 | 39.22 | 46.18 | 39.19 | 46.27 |
| | SI-SNR | 60.13 | 38.55 | 45.99 | 38.35 | 45.75 |
| | **SDSC(Ours)** | 61.64 | 39.75 | 47.38 | 40.34 | 47.28 |
| | **Hybrid(Ours)** | 62.00 | 39.06 | 47.75 | 41.45 | 47.70 |

Table 5, Appendix A.11 report the results with frozen encoders, while Table 6, Appendix A.12 summarize the performance with fine-tuning. When encoders are frozen, pre-trained SDSC models consistently outperform others in in-domain settings. In other settings, performance differences depend on the characteristics of each dataset. For instance, the epilepsy dataset relies heavily on amplitude patterns, where pre-trained MSE models perform better. In contrast, the gesture dataset depends on the waveform structure, and SDSC models consistently achieve higher accuracy. These results highlight the need to select the reconstruction objective according to the properties of the signal. Pre-trained hybrid models provide a consistent choice across datasets, maintaining balanced performance. Lastly, SDSC models consistently achieve higher precision, suggesting that structural reconstruction contributes to more semantically meaningful representations in signal-based classification.

Table 6: Summary of fine-tuning classification performance, averaged across scenarios. (full results in Appendix A.12.)

| Scenario | Loss | Acc.↑ | Prec.↑ | Rec.↑ | F1↑ | Avg↑ |
|---|---|---|---|---|---|---|
| **Avg (In Domain)** | MSE | 79.66 | 75.87 | 71.24 | 71.09 | 74.46 |
| | Soft-DTW | 79.07 | 75.68 | 70.01 | 69.43 | 73.55 |
| | PCC | **79.76** | **76.17** | **71.34** | **71.21** | **74.62** |
| | SI-SNR | 79.31 | 75.85 | 70.27 | 70.31 | 73.94 |
| | **SDSC(Ours)** | 79.60 | 76.01 | 70.54 | 70.69 | 74.21 |
| | **Hybrid(Ours)** | 79.52 | 75.60 | 71.10 | 70.21 | 74.11 |
| **Avg (Cross Domain)** | MSE | 83.74 | 85.46 | **84.89** | **84.33** | **84.65** |
| | Soft-DTW | 83.20 | 84.40 | 84.55 | 83.89 | 84.05 |
| | PCC | 83.79 | 86.78 | 82.37 | 83.03 | 83.99 |
| | SI-SNR | **84.27** | 84.75 | 84.13 | 83.19 | 84.09 |
| | **SDSC(Ours)** | 83.27 | **85.93** | 81.33 | 82.62 | 83.29 |
| | **Hybrid(Ours)** | 83.23 | 85.09 | 83.35 | 82.98 | 83.66 |

## 5 CONCLUSIONS

In this paper, we introduced the Signal Dice Similarity Coefficient (SDSC), a structure-aware reconstruction metric for semantic representation learning in time-series data. In this work, structure-aware refers to local waveform consistency characterized by sign and magnitude overlap, rather than global alignment or phase warping. Unlike traditional distance-based metrics, SDSC is defined over a normalized range of $[0, 1]$ and is robust to amplitude variations. Since SDSC is not directly differentiable, we proposed a smooth approximation to enable its use as a loss. We also identified limitations in amplitude-sensitive tasks, where SDSC may overlook critical signal properties. To address this, we proposed a hybrid loss that combines SDSC with MSE.

Our results demonstrate that SDSC improves representation quality under identical contrastive settings by enhancing local structural fidelity in the reconstruction branch. Although the improvements are moderate, they confirm that structure-aware reconstruction contributes complementary information to contrastive learning.It is also worth noting that the comparable downstream performance between MSE and SDSC does not necessarily imply the superiority of MSE. Rather, it suggests that amplitude-based metrics like MSE may overestimate reconstruction quality by assigning low errors even to structurally inconsistent signals. SDSC, in contrast, exposes such cases more transparently, revealing the true limits of purely distance-based objectives. This confirms that the two objectives are compatible and can coexist without interference. Our key findings revealed that while SDSC-based models often showed higher reconstruction errors in terms of MSE, they achieved comparable downstream forecasting task performance under identical contrastive settings. This suggests that excessive MSE minimization provides diminishing returns and that the structural alignment captured by SDSC is a sufficient, and perhaps more efficient, objective to learning effective representations. Furthermore, in classification tasks, SDSC consistently improved performance in in-domain settings when encoders were frozen, demonstrating SDSC's ability to preserve semantic structure. While alignment-based objectives such as SoftDTW or DILATE remain stronger baselines in certain forecasting settings, their quadratic complexity makes them impractical at scale. Our complexity analysis highlights SDSC as a lightweight, alignment-free alternative that achieves comparable downstream performance at a fraction of the computational cost.

In conclusion, our results question the default reliance on MSE in signal pre-training and position SDSC as a promising metric for structure-aware learning in time-series domains.Future work may investigate the integration of SDSC in other self-supervised frameworks and its effect on domain adaptation or cross-modal tasks. In addition, understanding the relationship between structural similarity and task-specific generalization remains an open question. We leave head-to-head training with SoftDTW/DILATE and integration into additional pretraining frameworks (e.g., TI-MAE/contrastive) as future work, noting compute constraints. Finally, we provide a practical guideline in A.14 summarizing when SDSC, MSE, or the hybrid loss is preferred under different learning regimes.

## 6 REPRODUCIBILITY STATEMENT

All source code for our experiments is available in an anonymous GitHub repository to ensure full reproducibility. This repository includes the implementation of our proposed SDSC loss function and the scripts to reproduce all forecasting and classification results presented in Section 4. The key hyperparameters for all pre-training and fine-tuning experiments are detailed in Appendix A.3. The mathematical derivation of SDSC is provided in Section 3.2, and a formal proof of its boundedness is available in Appendix B. $https : //anonymous.4open.science/r/Signal_Dice_similarity_Coefficient - 75A4$

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

# A  APPENDIX

## A.1  COMPUTATIONAL COMPLEXITY ANALYSIS

Alignment-based objectives such as DTW, FastDTW, SoftDTW, and DILATE have been widely used to handle temporal misalignments in time-series. However, these approaches are fundamentally distance-based and typically quadratic in sequence length $T$, which can be prohibitive for large-scale pre-training. FastDTW reduces the complexity to linear time but remains approximate and non-differentiable, limiting its applicability for gradient-based optimization. SoftDTW provides a differentiable relaxation but still incurs $O(T^2)$ complexity. DILATE combines shape and temporal distortion terms, yet inherits the same quadratic cost and has been applied only in forecasting tasks.

Table 7: Comparison of structure-aware metrics for time-series. $T$ denotes sequence length. SDSC differs by being alignment-free, lightweight, and interpretable.

| Method | Type | Time Complexity | Remarks |
|---|---|---|---|
| PCC | Correlation-based | $O(T)$ | Measures linear dependence, sensitive to phase shift |
| SI-SNR | Ratio-based | $O(T)$ | Structure-aware in audio, indirect via signal-to-error ratio |
| DTW | Distance-based | $O(T^2)$ | Exact alignment, non-differentiable |
| FastDTW | Distance-based | $O(T)$ (approx.) | Efficient but approximate, non-differentiable |
| SoftDTW | Distance-based | $O(T^2)$ | Differentiable relaxation of DTW |
| DILATE | Distance-based | $O(T^2)$ | Combines shape and temporal distortion tailored for forecasting |
| **SDSC (ours)** | Similarity-based | $O(T)$ | Overlap-based, bounded in [0,1], differentiable via sigmoid |

In contrast, our proposed SDSC is an alignment-free similarity measure. It operates in linear time $O(T)$, is bounded in $[0, 1]$, and can be smoothly approximated for differentiation, making it lightweight and practical for representation learning at scale. Table 7 summarizes the key properties of these methods.

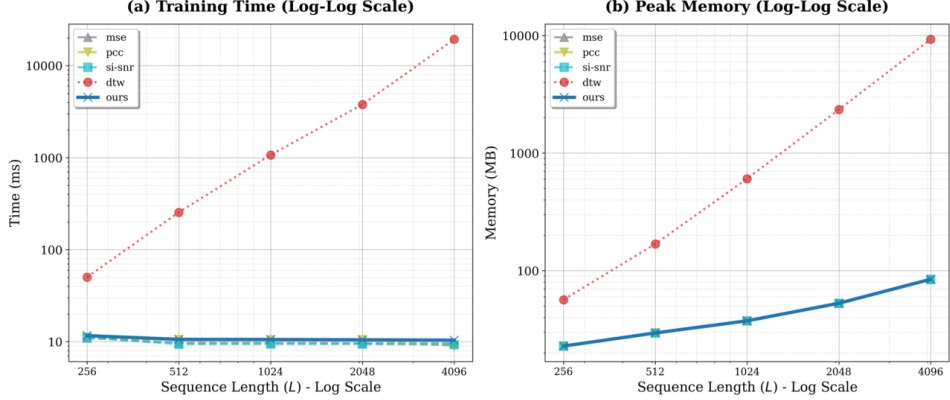

Figure 4: Computational efficiency and scalability comparison across varying sequence lengths ($L$) (log-log scale). While Soft-DTW demonstrates quadratic scaling ($O(L^2)$) with significantly higher time and memory costs, the proposed method maintains linear scalability ($O(L)$), achieving efficiency comparable to simple point-wise losses like MSE, PCC, and SI-SNR. Results are averaged over 50 training steps with a fixed batch size.

## A.2 HEAVISIDE CONVENTION

$$H(x) = \begin{cases} 1, & x > 0, \\ 0, & x \leq 0. \end{cases} \tag{10}$$

This convention sets $H(0) = 0$, which we adopt as the default throughout both evaluation and training. During optimization, we use a smooth sigmoid relaxation $\hat{H}(x) = \frac{1}{1+e^{-\alpha x}}$ with $\alpha = 10$, so that $\hat{H}(x) \approx H(x)$ while remaining differentiable.

## A.3 GRADIENT SENSITIVITY ANALYSIS

In this section, we measure the gradient norm with respect to different types of signal perturbations to assess how each loss function responds to structural variations. The analysis is based on the same toy cases as in Table 1, including the jitter case.

Table 8: Analysis of gradient sensitivity. (a) Comparison across different loss functions. (b) Effect of the sharpness parameter $\alpha$ for the SDSC loss.

<table>
<tr><td colspan="4" align="center">(a) Gradient Sensitivity</td><td colspan="4" align="center">(b) Sensitivity to $\alpha$</td></tr>
<tr><th>Example</th><th>MSE</th><th>MAE</th><th>SDSC</th><th>Example</th><th>$\alpha = 1$</th><th>$\alpha = 10$</th><th>$\alpha = 100$</th></tr>
<tr><td>Inverted</td><td>0.0894</td><td>0.0316</td><td>0.0000</td><td>Inverted</td><td>0.0091</td><td>0.0082</td><td>0.0047</td></tr>
<tr><td>0.5x Scaled</td><td>0.0223</td><td>0.0316</td><td>0.0442</td><td>0.5x Scaled</td><td>0.0289</td><td>0.0437</td><td>0.0436</td></tr>
<tr><td>2x Scaled</td><td>0.0447</td><td>0.0316</td><td>0.0110</td><td>2x Scaled</td><td>0.0062</td><td>0.0102</td><td>0.0102</td></tr>
<tr><td>Zero</td><td>0.0447</td><td>0.0316</td><td>0.0000</td><td>Zero</td><td>0.0000</td><td>0.0000</td><td>0.0000</td></tr>
<tr><td>Noise Sample</td><td>0.0194</td><td>0.0316</td><td>0.0237</td><td>Noise Sample</td><td>0.0152</td><td>0.0228</td><td>0.0237</td></tr>
<tr><td>Shifted</td><td>0.0632</td><td>0.0316</td><td>0.0075</td><td>Shifted</td><td>0.0074</td><td>0.0087</td><td>0.0076</td></tr>
<tr><td>Jittered</td><td>0.0032</td><td>0.0316</td><td>0.0248</td><td>Jittered</td><td>0.0165</td><td>0.0228</td><td>0.0242</td></tr>
</table>

Tables 8a and 8b present the gradient norms for MSE, MAE, and SDSC, and the effect of varying the $\alpha$ parameter in SDSC, respectively.

MSE exhibits significant gradient changes under amplitude perturbations, but it fails to respond meaningfully to minor structural variations, such as jitter. In contrast, SDSC yields low gradients for structure-preserving signals (e.g., shifted), while responding with larger gradients when local waveform patterns are distorted (e.g., jittered). Although gradient vanishing may occur in some structural-breaking cases, SDSC remains robust to amplitude shifts.

As $\alpha$ increases, the behavior of the approximated SDSC becomes more similar to that of the original formulation. The results suggest that $\alpha = 10$ is sufficient for a close approximation in practice.

### A.3.1 DOWNSTREAM TASK ANALYSIS WITH $\alpha$

Table 9: Impact of the sharpness parameter $\alpha$ on pre-training reconstruction performance across scenarios.

| Scenario | Example | MSE↓ | MAE↓ | SDSC↑ |
|---|---|---|---|---|
| Forecasting | $\alpha = 1$ | 1.7030 | 0.9112 | 0.3946 |
| | $\alpha = 10$ | 1.7230 | 0.9140 | 0.3975 |
| | $\alpha = 100$ | 1.7410 | 0.9177 | 0.3995 |
| Classification | $\alpha = 1$ | 0.1012 | 0.2406 | 0.6808 |
| | $\alpha = 10$ | 0.1069 | 0.2471 | 0.6794 |
| | $\alpha = 100$ | 0.1090 | 0.2468 | 0.6869 |

Table 10: Downstream task performance comparison under different $\alpha$ values. (a) Forecasting error metrics (lower is better). (b) Classification performance metrics (higher is better).

<table>
<tr><td colspan="3" align="center">(a) Forecasting</td><td colspan="6" align="center">(b) Classification</td></tr>
<tr><th>Example</th><th>MSE↓</th><th>MAE↓</th><th>Example</th><th>Acc.↑</th><th>Prec.↑</th><th>Rec.↑</th><th>F1↑</th><th>Avg↑</th></tr>
<tr><td>$\alpha = 1$</td><td>1.7030</td><td>0.9112</td><td>$\alpha = 1$</td><td>92.38</td><td>94.67</td><td>84.04</td><td>87.82</td><td>89.73</td></tr>
<tr><td>$\alpha = 10$</td><td>1.7230</td><td>0.9140</td><td>$\alpha = 10$</td><td>93.09</td><td>94.56</td><td>83.44</td><td>87.65</td><td>89.69</td></tr>
<tr><td>$\alpha = 100$</td><td>1.7410</td><td>0.9177</td><td>$\alpha = 100$</td><td>93.28</td><td>94.16</td><td>84.30</td><td>88.15</td><td>89.97</td></tr>
</table>

## A.4 HYPERPARAMETERS

Preprocessing. We apply z-score normalization using a StandardScaler fit only on the training split, and reuse its statistics to transform validation and test splits (preventing leakage). For multivariate inputs, we normalize per channel. For UEA datasets we follow the provided Normalizer. All normalization is applied before computing reconstruction losses.

The following tables summarize the key hyperparameters used for our pre-training and fine-tuning experiments.

### A.4.1 FORECASTING HYPERPARAMETERS

Table 11: Pre-training

| Parameter | Value |
| --- | --- |
| Sequence Length | 96 |
| Learning rate | $1 \times 10^{-4}$ |
| Batch size | 16 |
| Epochs | 50 |
| Alpha (for SDSC) | 10 |
| Optimizer | Adam |

Table 12: Fine-tuning

| Parameter | Value |
| --- | --- |
| Sequence Length | 96 |
| Learning rate | $1 \times 10^{-4}$ |
| Batch size | 16 |
| Epochs | 10 |
| Alpha (for SDSC) | 10 |
| Optimizer | Adam |

### A.4.2 CLASSIFICATION HYPERPARAMETERS

Table 13: Pre-training

| Parameter | Value |
| --- | --- |
| Learning rate | $1 \times 10^{-4}$ |
| Batch size | 32 |
| Epochs | 20 |
| Alpha (for SDSC) | 10 |
| Optimizer | Adam |

Table 14: Fine-tuning

| Parameter | Value |
| --- | --- |
| Learning rate | $1 \times 10^{-4}/3 \times 10^{-4}$ |
| Batch size | 32 |
| Epochs | 100/300 |
| Alpha (for SDSC) | 10 |
| Optimizer | Adam |

## A.5 FULL PRE-TRAINING FORECASTING RESULTS

Table 15: Full pre-training reconstruction performance for forecasting datasets.

| Dataset | Loss | MSE↓ | MAE↓ | SDSC↑ |
|---|---|---|---|---|
| ETTh1 | MSE | 1.1980 | 0.6863 | 0.5573 |
| | SoftDTW | 1.9270 | 0.9424 | 0.4199 |
| | PCC | 2.4280 | 1.0950 | 0.4051 |
| | SI-SNR | 221.9000 | 11.6300 | 0.0713 |
| | **SDSC** | 1.5430 | 0.7653 | 0.5725 |
| | **Hybrid** | 1.2070 | 0.6738 | 0.5774 |
| ETTh2 | MSE | 0.6229 | 0.4595 | 0.7263 |
| | SoftDTW | 0.9799 | 0.6386 | 0.6229 |
| | PCC | 0.9711 | 0.6350 | 0.6190 |
| | SI-SNR | 0.9586 | 0.6263 | 0.6242 |
| | **SDSC** | 0.6553 | 0.4643 | 0.7206 |
| | **Hybrid** | 0.6210 | 0.4487 | 0.7320 |
| ETTm1 | MSE | 1.1740 | 0.6532 | 0.5759 |
| | SoftDTW | 1.8140 | 0.8955 | 0.4334 |
| | PCC | 2.6570 | 1.1030 | 0.4149 |
| | SI-SNR | 1.7990 | 0.9406 | 0.3999 |
| | **SDSC** | 1.7880 | 0.7975 | 0.5838 |
| | **Hybrid** | 1.1780 | 0.6420 | 0.5886 |
| ETTm2 | MSE | 0.0269 | 0.1121 | 0.9355 |
| | SoftDTW | 0.7497 | 0.5150 | 0.7078 |
| | PCC | 0.7235 | 0.5016 | 0.7105 |
| | SI-SNR | 0.7413 | 0.4978 | 0.7155 |
| | **SDSC** | 0.0280 | 0.1138 | 0.9345 |
| | **Hybrid** | 0.0242 | 0.1052 | 0.9395 |
| Weather | MSE | 0.0863 | 0.1126 | 0.9022 |
| | SoftDTW | 0.7056 | 0.4354 | 0.5886 |
| | PCC | 0.6908 | 0.4337 | 0.5970 |
| | SI-SNR | 0.6848 | 0.4149 | 0.6189 |
| | **SDSC** | 0.1024 | 0.1045 | 0.9097 |
| | **Hybrid** | 0.0806 | 0.1007 | 0.9132 |
| Electricity | MSE | 0.0970 | 0.2136 | 0.8544 |
| | SoftDTW | 1.1060 | 0.7871 | 0.4252 |
| | PCC | 0.7070 | 0.6366 | 0.5180 |
| | SI-SNR | 1.0260 | 0.6743 | 0.5398 |
| | **SDSC** | 0.1039 | 0.2215 | 0.8524 |
| | **Hybrid** | 0.0842 | 0.1960 | 0.8679 |
| Traffic | MSE | 0.1914 | 0.2304 | 0.8382 |
| | SoftDTW | 2.0090 | 0.9881 | 0.2949 |
| | PCC | 1.1250 | 0.7226 | 0.4270 |
| | SI-SNR | 17.2500 | 3.0020 | 0.1964 |
| | **SDSC** | 0.2229 | 0.2418 | 0.8327 |
| | **Hybrid** | 0.1529 | 0.1914 | 0.8700 |
| **Avg (Forecasting)** | MSE | 0.4852 | 0.3525 | 0.7670 |
| | SoftDTW | 1.3273 | 0.7432 | 0.4990 |
| | PCC | 1.3289 | 0.6705 | 0.5274 |
| | SI-SNR | 34.9085 | 2.5408 | 0.4523 |
| | **SDSC** | 0.6348 | 0.3870 | 0.7723 |
| | **Hybrid** | **0.4783** | **0.3368** | **0.7841** |

Note: SI-SNR values are reported for completeness, but they are on a different numerical scale than MSE/MAE/SDSC. In some datasets (e.g., ETTh1), training with SI-SNR fails to converge, leading to unstable or very large values. These results should therefore be interpreted with caution (see main text for discussion).

## A.6  FULL $\lambda = 0.5$ PRE-TRAINING FORECASTING RESULTS:

Table 16: Full pre-training reconstruction performance for forecasting datasets with fixed $\lambda = 0.5$.

| Dataset | MSE↓ | MAE↓ | SDSC↑ |
|---|---|---|---|
| ETTh1 | 1.7480 | 0.9185 | 0.3997 |
| ETTh2 | 0.9359 | 0.6218 | 0.6262 |
| ETTm1 | 1.8250 | 0.9354 | 0.4026 |
| ETTm2 | 0.7236 | 0.5068 | 0.7060 |
| Weather | 0.6919 | 0.4449 | 0.5721 |
| Electricity | 0.7146 | 0.6547 | 0.4703 |
| Traffic | 0.9691 | 0.6406 | 0.4929 |
| **Avg (Forecasting)** | 1.0869 | 0.6747 | 0.5243 |

## A.7 FULL PRE-TRAINING CLASSIFICATION RESULTS

Table 17: Full pre-training reconstruction performance for classification datasets.

| Dataset | Loss | MSE↓ | MAE↓ | SDSC↑ |
|---|---|---|---|---|
| Epilepsy | MSE | 0.0807 | 0.2456 | 0.6244 |
| | Soft-DTW | **0.0719** | **0.2384** | 0.6407 |
| | PCC | 0.3128 | 0.4543 | 0.2468 |
| | SI-SNR | 0.3235 | 0.4612 | 0.2435 |
| | **SDSC(Ours)** | 0.1300 | 0.2740 | 0.6736 |
| | **Hybrid(Ours)** | 0.1099 | 0.2484 | **0.6856** |
| SleepEEG | MSE | 100.5599 | 6.8082 | 0.5966 |
| | Soft-DTW | **98.1958** | **6.7118** | 0.6012 |
| | PCC | 239.7082 | 8.5638 | 0.0775 |
| | SI-SNR | 236.8986 | 8.5080 | 0.0950 |
| | **SDSC(Ours)** | 147.9205 | 7.4512 | **0.6483** |
| | **Hybrid(Ours)** | 100.5852 | 6.8087 | 0.6105 |
| **Avg (Classification)** | MSE | 50.3203 | 3.5269 | 0.6105 |
| | Soft-DTW | **49.1339** | **3.4751** | 0.6210 |
| | PCC | 120.0105 | 4.5091 | 0.1622 |
| | SI-SNR | 118.6110 | 4.4846 | 0.1693 |
| | **SDSC(Ours)** | 74.0253 | 3.8626 | **0.6610** |
| | **Hybrid(Ours)** | 50.3471 | 3.5286 | 0.6481 |

## A.8 FULL $\lambda = 0.5$ PRE-TRAINING CLASSIFICATION RESULTS:

Table 18: Full pre-training reconstruction performance for classification datasets with fixed $\lambda = 0.5$.

| Dataset | MSE↓ | MAE↓ | SDSC↑ |
|---|---|---|---|
| Epilepsy | 0.1096 | 0.2470 | 0.6881 |
| SleepEEG | 110.3683 | 7.1989 | 0.5623 |
| **Avg (Classification)** | 55.2390 | 3.7230 | 0.6252 |

## A.9 FULL FORECASTING RESULT

Table 19: Forecasting fine-tuning performance with all baselines.

| Dataset | Pre-training Loss | MSE↓ | MAE↓ |
|---|---|---|---|
| ETTh1 | MSE | 0.380 | 0.408 |
| | Soft-DTW | 0.384 | 0.410 |
| | PCC | 0.382 | 0.408 |
| | SI-SNR | 0.381 | **0.399** |
| | **SDSC(Ours)** | **0.379** | 0.406 |
| | **Hybrid(Ours)** | 0.382 | 0.406 |
| ETTh2 | MSE | 0.304 | 0.350 |
| | Soft-DTW | 0.306 | 0.351 |
| | PCC | 0.304 | 0.350 |
| | SI-SNR | **0.303** | **0.349** |
| | **SDSC(Ours)** | 0.306 | 0.352 |
| | **Hybrid(Ours)** | 0.304 | 0.350 |
| ETTm1 | MSE | 0.327 | **0.363** |
| | Soft-DTW | 0.327 | 0.365 |
| | PCC | 0.321 | 0.364 |
| | SI-SNR | 0.345 | 0.373 |
| | **SDSC(Ours)** | **0.324** | 0.364 |
| | **Hybrid(Ours)** | 0.325 | 0.364 |
| ETTm2 | MSE | **0.185** | **0.275** |
| | Soft-DTW | 0.190 | 0.278 |
| | PCC | 0.194 | 0.279 |
| | SI-SNR | 0.189 | 0.279 |
| | **SDSC(Ours)** | 0.191 | 0.278 |
| | **Hybrid(Ours)** | 0.188 | 0.276 |
| Weather | MSE | 0.169 | **0.213** |
| | Soft-DTW | 0.176 | 0.220 |
| | PCC | **0.168** | 0.215 |
| | SI-SNR | 0.196 | 0.239 |
| | **SDSC(Ours)** | **0.168** | **0.213** |
| | **Hybrid(Ours)** | 0.169 | 0.215 |
| Electricity | MSE | 0.200 | 0.291 |
| | Soft-DTW | 0.200 | 0.290 |
| | PCC | 0.199 | **0.288** |
| | SI-SNR | 0.203 | 0.292 |
| | **SDSC(Ours)** | 0.200 | 0.293 |
| | **Hybrid(Ours)** | **0.198** | **0.288** |
| Traffic | MSE | 0.497 | 0.312 |
| | Soft-DTW | 0.535 | 0.337 |
| | PCC | 0.505 | 0.319 |
| | SI-SNR | 0.550 | 0.345 |
| | **SDSC(Ours)** | **0.492** | **0.309** |
| | **Hybrid(Ours)** | 0.494 | 0.315 |
| Avg | MSE | 0.295 | **0.316** |
| | Soft-DTW | 0.303 | 0.322 |
| | PCC | 0.296 | 0.318 |
| | SI-SNR | 0.310 | 0.325 |
| | **SDSC(Ours)** | **0.294** | **0.316** |
| | **Hybrid(Ours)** | **0.294** | **0.316** |

## A.10 FULL $\lambda = 0.5$ FORECASTING RESULTS

Table 20: Forecasting fine-tuning performance with all baselines with fixed $\lambda = 0.5$.

| Dataset | MSE↓ | MAE↓ |
|---|---|---|
| ETTh1 | 0.380 | 0.405 |
| ETTh2 | 0.307 | 0.352 |
| ETTm1 | 0.319 | 0.362 |
| ETTm2 | 0.193 | 0.279 |
| Weather | 0.168 | 0.215 |
| Electricity | 0.200 | 0.289 |
| Traffic | 0.506 | 0.318 |
| **Avg (Forecasting)** | 0.2961 | 0.3171 |

## A.11 Full Freeze Classification Result

Table 21: Full freeze classification performance across all scenarios. Full results for all datasets are presented to ensure reproducibility.

| Scenario | | Loss | Acc.↑ | Prec.↑ | Rec.↑ | F1↑ | Avg↑ |
|---|---|---|---|---|---|---|---|
| **In Domain** | Epilepsy ↓ Epilepsy | MSE | 90.69 | 93.23 | 77.23 | 82.26 | 85.85 |
| | | Soft-DTW | 80.21 | 40.11 | 50.00 | 44.51 | 53.71 |
| | | PCC | 80.21 | 40.10 | 50.00 | 44.51 | 53.71 |
| | | SI-SNR | 80.21 | 40.10 | 50.00 | 44.51 | 53.71 |
| | | **SDSC(Ours)** | 91.86 | 93.82 | 80.27 | 84.98 | 87.73 |
| | | **Hybrid(Ours)** | **92.44** | **93.58** | **82.13** | **86.38** | **88.63** |
| | SleepEEG ↓ SleepEEG | MSE | 60.20 | 52.90 | 49.75 | 46.92 | 52.44 |
| | | Soft-DTW | 57.31 | 41.67 | 44.72 | 42.52 | 46.56 |
| | | PCC | **62.59** | 54.65 | 53.12 | 48.88 | 54.81 |
| | | SI-SNR | 61.95 | **54.69** | **53.62** | **49.30** | **54.89** |
| | | **SDSC(Ours)** | 60.89 | 54.55 | 49.59 | 46.72 | 52.94 |
| | | **Hybrid(Ours)** | 60.01 | 51.86 | 49.09 | 46.55 | 51.88 |
| **Cross Domain** | SleepEEG ↓ Epilepsy | MSE | 80.21 | 40.11 | 50.00 | 44.51 | 53.71 |
| | | Soft-DTW | 80.21 | 40.11 | 50.00 | 44.51 | 53.71 |
| | | PCC | 80.21 | 40.11 | 50.00 | 44.51 | 53.71 |
| | | SI-SNR | 80.21 | 40.11 | 50.00 | 44.51 | 53.71 |
| | | **SDSC(Ours)** | 80.21 | 40.11 | 50.00 | 44.51 | 53.71 |
| | | **Hybrid(Ours)** | 80.21 | 40.11 | 50.00 | 44.51 | 53.71 |
| | SleepEEG ↓ FD-B | MSE | **52.19** | 35.80 | 38.21 | 33.87 | 40.02 |
| | | Soft-DTW | 52.30 | 37.99 | **38.49** | **36.06** | **41.21** |
| | | PCC | 52.00 | 37.64 | 38.06 | 32.29 | 39.99 |
| | | SI-SNR | 49.79 | **39.44** | 36.45 | 28.84 | 38.63 |
| | | **SDSC(Ours)** | 51.71 | 37.11 | 37.86 | 31.84 | 39.63 |
| | | **Hybrid(Ours)** | 51.43 | 34.34 | 37.65 | 34.01 | 39.96 |
| | SleepEEG ↓ Gesture | MSE | **70.00** | 64.80 | **70.00** | **66.17** | 67.74 |
| | | Soft-DTW | 69.17 | **72.31** | 69.16 | 65.22 | **68.97** |
| | | PCC | 63.33 | 63.68 | 63.33 | 58.84 | 62.30 |
| | | SI-SNR | 64.16 | 59.18 | 64.16 | 58.92 | 61.61 |
| | | **SDSC(Ours)** | 68.33 | 66.34 | 68.33 | 63.89 | 66.72 |
| | | **Hybrid(Ours)** | **70.00** | 66.09 | **70.00** | **66.17** | 68.07 |
| | SleepEEG ↓ EMG | MSE | 46.34 | 15.45 | 33.33 | 21.11 | 29.06 |
| | | Soft-DTW | 46.34 | 15.45 | 33.33 | 21.11 | 29.0 |
| | | PCC | 46.34 | 15.45 | 33.33 | 21.11 | 29.06 |
| | | SI-SNR | 46.34 | 15.45 | 33.33 | 21.11 | 29.06 |
| | | **SDSC(Ours)** | 46.34 | 15.45 | 33.33 | 21.11 | 29.06 |
| | | **Hybrid(Ours)** | 46.34 | 15.45 | 33.33 | 21.11 | 29.06 |

In some cross-domain cases (e.g., SleepEEG→Epilepsy, SleepEEG→EMG), the reported metrics are identical across losses. This is not due to code reuse but rather because the small dataset failed to converge under all objectives, leading to degenerate identical predictions. We verified by running independent trials.

## A.12 FULL CLASSIFICATION RESULT

Table 22: Full fine-tuning classification performance across all scenarios.

| Scenario | | Loss | Acc.↑ | Prec.↑ | Rec.↑ | F1↑ | Avg↑ |
|---|---|---|---|---|---|---|---|
| **In Domain** | Epilepsy ↓ Epilepsy | MSE | **94.20** | **94.64** | **86.77** | **90.02** | **91.40** |
| | | Soft-DTW | 93.08 | 94.16 | 83.69 | 87.71 | 89.66 |
| | | PCC | 94.07 | **94.64** | 86.34 | 89.74 | 91.20 |
| | | SI-SNR | 93.15 | 94.18 | 83.88 | 87.86 | 89.78 |
| | | **SDSC(Ours)** | 93.97 | 94.46 | 86.16 | 89.56 | 91.04 |
| | | **Hybrid(Ours)** | 93.91 | 94.59 | 85.89 | 89.42 | 90.95 |
| | SleepEEG ↓ SleepEEG | MSE | 65.11 | 57.10 | 55.70 | 52.16 | 57.52 |
| | | Soft-DTW | 65.05 | 57.19 | 56.32 | 51.15 | 57.43 |
| | | PCC | 65.44 | **57.69** | 56.34 | 52.68 | 58.04 |
| | | SI-SNR | **65.46** | 57.51 | 56.66 | **52.75** | **58.09** |
| | | **SDSC(Ours)** | 65.22 | 57.55 | 54.92 | 51.81 | 57.38 |
| | | **Hybrid(Ours)** | 65.13 | 56.62 | 56.30 | 51.00 | 57.26 |
| **Cross Domain** | SleepEEG ↓ Epilepsy | MSE | 95.19 | 94.30 | 90.20 | 92.07 | 92.94 |
| | | Soft-DTW | **95.40** | 94.85 | **90.35** | **92.39** | **93.25** |
| | | PCC | 91.35 | 94.25 | 78.57 | 83.65 | 86.96 |
| | | SI-SNR | 94.16 | **95.24** | 86.16 | 89.82 | 91.35 |
| | | **SDSC(Ours)** | 92.03 | 94.33 | 80.48 | 85.27 | 88.03 |
| | | **Hybrid(Ours)** | **95.19** | 94.30 | **90.20** | **92.07** | **92.94** |
| | SleepEEG ↓ FD-B | MSE | 63.88 | 69.36 | 72.98 | 69.98 | 69.05 |
| | | Soft-DTW | 60.68 | 65.76 | 70.64 | 67.68 | 66.19 |
| | | PCC | **66.24** | **73.78** | 72.88 | 71.85 | **71.24** |
| | | SI-SNR | 65.35 | 65.10 | 72.30 | 66.48 | 67.31 |
| | | **SDSC(Ours)** | 65.94 | 72.33 | 73.45 | **72.29** | 71.00 |
| | | **Hybrid(Ours)** | 64.26 | 68.02 | **73.50** | 70.06 | 68.96 |
| | SleepEEG ↓ Gesture | MSE | 78.33 | 79.85 | 78.33 | 77.13 | 78.41 |
| | | Soft-DTW | 79.17 | 78.65 | 79.17 | 77.35 | 78.58 |
| | | PCC | **80.00** | 80.77 | **80.00** | 78.47 | 79.81 |
| | | SI-SNR | **80.00** | 80.33 | **80.00** | 78.31 | 79.66 |
| | | **SDSC(Ours)** | **80.00** | 80.21 | **80.00** | 79.32 | **79.88** |
| | | **Hybrid(Ours)** | 78.33 | **81.20** | 78.33 | 76.15 | 78.50 |
| | SleepEEG ↓ EMG | MSE | **97.56** | **98.33** | **98.04** | **98.14** | **98.18** |
| | | Soft-DTW | **97.56** | **98.33** | **98.04** | **98.14** | **98.18** |
| | | PCC | **97.56** | **98.33** | **98.04** | **98.14** | **98.18** |
| | | SI-SNR | **97.56** | **98.33** | **98.04** | **98.14** | **98.18** |
| | | **SDSC(Ours)** | 95.12 | 96.83 | 91.37 | 93.62 | 94.24 |
| | | **Hybrid(Ours)** | 95.12 | 96.83 | 91.37 | 93.62 | 94.24 |

A.13   FULL $\lambda = 0.5$ CLASSIFICATION RESULTS

Table 23: Full freeze classification performance across all scenarios with fixed $\lambda = 0.5$. Full results for all datasets are presented to ensure reproducibility.

| Scenario | | Acc.↑ | Prec.↑ | Rec.↑ | F1↑ | Avg↑ |
|---|---|---|---|---|---|---|
| **In Domain** | Epilepsy ↓ Epilepsy | 80.21 | 40.11 | 50.00 | 44.51 | 46.21 |
| | SleepEEG ↓ SleepEEG | 58.78 | 50.94 | 46.69 | 44.31 | 50.18 |
| **Cross Domain** | SleepEEG ↓ Epilepsy | 80.21 | 40.11 | 50.00 | 44.51 | 46.21 |
| | SleepEEG ↓ FD-B | 50.02 | 36.61 | 38.09 | 34.51 | 39.81 |
| | SleepEEG ↓ Gesture | 68.33 | 68.03 | 68.33 | 64.60 | 67.32 |
| | SleepEEG ↓ EMG | 46.34 | 15.45 | 33.33 | 21.11 | 29.06 |

In some cross-domain cases (e.g., SleepEEG→Epilepsy, SleepEEG→EMG), the reported metrics are identical across losses. This is not due to code reuse but rather because the small dataset failed to converge under all objectives, leading to degenerate identical predictions. We verified by running independent trials.

## A.14 Practical Guideline for Loss Selection

| Loss Type | Recommended Usage Scenario | Example Dataset / Task |
|---|---|---|
| $\mathcal{L}_{\text{MSE}}$ | Amplitude-critical tasks requiring precise numeric matching; performs well when signal magnitude is stable across domains. | FD-B (Cross-domain forecasting) |
| $\mathcal{L}_{\text{SDSC}}$ | Structure-critical tasks emphasizing polarity and waveform shape consistency; suitable for oscillatory or physiological signals. | SleepEEG (In-domain SSL pretraining) |
| Hybrid ($\lambda_{sdsc}$,$\lambda_{mse}$) | Balanced reconstruction for tasks with mixed structural and amplitude sensitivity; provides stable representation across varied regimes. | Epilepsy EEG (Hybrid reconstruction and fine-tuning) |

Table 24: Guidelines for selecting between MSE, SDSC, and the hybrid loss across different scenarios and datasets.

## A.15 The Use of Large Language Models

We utilized a large language model (e.g., Google's Gemini, OpenAI's GPT-4) to assist in polishing the writing of the manuscript. Its role was limited to improving clarity, refining grammar, and ensuring consistent terminology. The core ideas, experimental design, and all results and analyses presented in this paper are entirely our own.

# B  LEMMA

**Lemma 1** (Boundedness of SDSC). *For any two discrete signals $E = \{E(s)\}_{s \in S}$ and $R = \{R(s)\}_{s \in S}$, the Signal Dice Similarity Coefficient, SDSC($E, R$), is bounded such that $0 \leq SDSC \leq 1$.*

*Proof.* We begin with the definition of the discrete SDSC:

$$SDSC(E(t), R(t)) \approx \frac{2 \cdot \sum(H(S(s)) \cdot M(s))}{\sum(|E(s)| + |R(s)|) + \epsilon}$$

Here, $H(\cdot)$ is the Heaviside step function, $s \in S$ and $S$ is the set of discrete sampling points with $S \subset T$, and $\epsilon$ is a small constant to prevent division by zero.

**Lower Bound (SDSC $\geq 0$).**  The terms in the numerator and denominator are analyzed for non-negativity at each sampling point $s \in S$:

- The Heaviside function, $H(E(s)R(s))$, has a range of $\{0, 1\}$, so it is always non-negative.

- The term $M(s)$ is a minimum of absolute values, so it is always non-negative.

- The terms in the denominator, $|E(s)|$ and $|R(s)|$, are absolute values and thus always non-negative.

Since the numerator is a sum of non-negative values and the denominator is also a sum of non-negative values, the entire fraction must be non-negative. Therefore, SDSC $\geq 0$.

**Upper Bound (SDSC $\leq 1$).**  To prove the upper bound, we show that for each sampling point $s \in S$, the corresponding term in the numerator's sum is less than or equal to the corresponding term in the denominator's sum. Let $N_s$ be the term from the numerator's sum and $D_s$ be the term from the denominator's sum:

$$N_s = 2 \cdot H(E(s)R(s)) \cdot M(s)$$
$$D_s = |E(s)| + |R(s)|$$

We consider two cases based on the signs of $E(s)$ and $R(s)$:

- **Case 1: Same sign or one is zero** ($E(s)R(s) \geq 0$). In this case, $H(E(s)R(s)) = 1$. The numerator term becomes $N_s = 2 \cdot M(s)$. For any two non-negative numbers $a, b$, the inequality $2 \cdot \min(a, b) \leq a + b$ always holds. Therefore, $N_s \leq D_s$.

- **Case 2: Different signs** ($E(s)R(s) < 0$). In this case, $H(E(s)R(s)) = 0$. The numerator term becomes $N_s = 0$. Since $D_s = |E(s)| + |R(s)| \geq 0$, the inequality $N_s \leq D_s$ clearly holds.

In all possible cases, we have shown that $N_s \leq D_s$ for every sampling point $s$. Since this is true for each term, the sum of the numerator terms must be less than or equal to the sum of the denominator terms:

$$\sum_{s \in S} 2 \cdot H(E(s)R(s)) \cdot M(s) \leq \sum_{s \in S}(|E(s)| + |R(s)|) \leq \sum_{s \in S}(|E(s)| + |R(s)|) + \epsilon$$

Therefore, the entire fraction must be less than or equal to 1, which means SDSC $\leq 1$.

Combining the two bounds, we have proven that $0 \leq SDSC \leq 1$. $\qquad\square$

