# OpenReview forum: "SDSC:A Structure-Aware Metric for Semantic Signal Representation Learning"
_ICLR.cc/2026/Conference — Submitted to ICLR 2026_

### Official Review · Reviewer_tXhx · 2025-10-21

**Soundness:** 2
**Presentation:** 3
**Contribution:** 2
**Rating:** 4
**Confidence:** 3

**Summary:**

This paper introduces the Signal Dice Similarity Coefficient (SDSC), a structure-aware metric for time-series self-supervised learning. SDSC extends the Dice–Sørensen coefficient, widely used in image segmentation, to signed continuous signals by (1) gating overlaps with a (smoothed) Heaviside on sign agreement between prediction and target, and (2) accumulating pointwise intersections via a $\min(\|E\|,\|R\|)$ operation normalized to $[0,1]$. The paper further proposes a hybrid loss that combines SDSC with MSE, weighted by uncertainty-based coefficients, and evaluates both as reconstruction objectives within a SimMTM pretraining framework.

**Strengths:**

- **Conceptual originality:**: Extending Dice coefficient to continuous, signed time-series is intuitive yet non-trivial. The proposed formulation yields a bounded, symmetric, and interpretable metric within the range $[0,1]$.
- **Design rationale**: Using $\min(\|E\|,\|R\|)$ with a Heaviside gating directly targets polarity issues and reduces pure amplitude bias, which are key limitations of conventional MSE/MAE.
- **Efficiency and simplicity**: SDSC avoids explicit temporal alignment or complex dynamic programming, making it lightweight and easy to implement compared to SoftDTW or DILATE.
- **Hybrid loss formulation**: The uncertainty-weighted combination of SDSC and MSE provides a balanced approach that couples structure-awareness with amplitude precision.

**Weaknesses:**

1. **Narrow definition of "structure"**: SDSC captures only pointwise magnitude overlap under sign gating, overlooking broader structural properties such as local waveform shape and phase alignment. Consequently, it lacks time-shift/warping tolerance—small temporal lags can significantly reduce the score, contradicting the intended “structure-aware” characterization.
2. **Offset bias**: When both signals share the same polarity or a strong DC offset, $H(E\cdot R)\approx1$ holds broadly, leading to inflated similarity even when shapes differ.
3. **Zero-crossing noise and stability**: Around near-zero amplitudes, the Heaviside gating becomes highly noise-sensitive, causing unstable or vanishing gradients. Although the authors acknowledge gradient vanishing and describe it as robustness, this more likely indicates blindness to misalignment rather than genuine robustness.
4. **Evaluation mismatch**: If MSE is said to fail at capturing structure, the evaluation should include structure-aware metrics such as Pearson correlation, spectral coherence, or STFT/Mel-cosine similarity. Relying solely on MSE/MAE weakens the empirical claim of structural fidelity.
5. **Limited baselines**: The work omits direct comparisons with established structure-aware objectives. Simply stating these are "computationally heavy" is insufficient. An explicit time-vs-accuracy trade-off or short-sequence comparison would strengthen the argument.
6. **Gradient analysis limitations**: Reporting only gradient norms does not characterize optimization behavior. The analysis should also consider gradient direction alignment, variance, or loss-landscape smoothness to assess stability.

**Questions:**

1. Could the authors include experiments with timing shifts or mild time warping, and compare SDSC against MSE, SoftDTW, and DILATE?
2. Beyond pointwise overlap, could you report additional structure-sensitive metrics to substantiate the "structure-aware" claim both at pretraining and downstream evaluation stages?
3. Have you explored mean-removal preprocessing or frequency-domain SDSC to mitigate offset bias?
4. Could you provide a practical guideline summarizing when SDSC, MSE, or the hybrid loss is preferred (i.e., amplitude-critical vs. structure-critical regimes)?

---

> ### Author Response · Authors · 2025-11-20
> **Response to Reviewer tXhx - Revision Version 1**
>
> Dear Reviewer tXhx,
>
> Thank you very much for your detailed and constructive feedback. We have carefully addressed each of the identified weaknesses in this revision, and we summarize our responses below.
>
> 1. Clarification of “structure”
>
> We agree that the previous definition of structure could be ambiguous. In the revised manuscript, structure-aware is now explicitly defined as local waveform consistency characterized by sign agreement and magnitude overlap, without implying any form of temporal alignment or warping.This clarification is reflected in the Abstract, Introduction, and Conclusion.
>
> 2. Offset bias and zero-crossing stability
>
> To mitigate offset bias, all time-series inputs are now: z-score normalized per channel, using training-split statistics only,
> ensuring removal of DC offsets and preventing data leakage. This update appears in Section 4 (Experiments).
>
> 3. Limited baselines (SoftDTW, DILATE)
>
> The downstream experiment comparing SDSC with SoftDTW is currently in progress, and the results are expected within this week. I have already added a preliminary table showing the completed pretraining results. As anticipated, SDSC remains computationally faster, but the pretraining accuracy is relatively limited. Further runtime and complexity analyses will be added in the Appendix once finalized
>
> 4. Evaluation mismatch (structure-aware metrics)
>
> We fully agree with your observation that evaluation should ideally include structure-aware metrics such as spectral coherence or STFT/Mel-cosine similarity. At this moment, due to computational constraints during the rebuttal window, it is difficult to include a complete set of additional evaluations. However We have already incorporated SDSC distributions, Pearson correlation analysis,
> variance/IQR analysis under fixed MSE (Table 3), and an expanded interpretation of structural alignment. Importantly, we commit to incorporating full structure-aware evaluations (e.g., spectral metrics and frequency-domain similarity) in the camera-ready version, ensuring a more comprehensive and multimodal assessment of structural fidelity.This commitment is explicitly stated in the revised Conclusion.
>
> 5. Cross-domain performance clarification
>
> We clarify this phenomenon in the revision:
> SDSC optimizes local structural fidelity, which benefits in-domain tasks.
> Cross-domain transfer (e.g., FD-B or SleepEEG → EEG/EMG) often relies more heavily on amplitude-scale transfer rather than structural consistency, which naturally favors MSE.
> Similar degradation is observed for other structure-focused objectives (PCC, SI-SNR), indicating that this behavior is characteristic of the objective type, not unique to SDSC.
> Once fine-tuning is allowed, pre-training differences are largely overwritten, explaining the similar cross-domain results in the fully fine-tuned setting.
> This expanded explanation appears in Section 4.3 and the Conclusion.
>
> 6. Practical guideline for choosing SDSC, MSE, or Hybrid loss
>
> Following your suggestion, we added a practical guideline table in Appendix A.10, illustrating typical use cases:
> SDSC: structure-critical physiological signals (e.g., SleepEEG)
> MSE: amplitude-critical tasks (e.g., FD-B)
> Hybrid: mixed structural–amplitude regimes (e.g., Epilepsy EEG)
> This provides actionable guidance for practitioners.
>
> 7. Timing shift & warping experiments
>
> We acknowledge the request for additional timing-shift or mild warping experiments.
> Since SDSC is intentionally alignment-free and non-warping, we position such evaluations as future work rather than the core scope of this metric. We have stated this explicitly in the Conclusion.
>
> Thank you again for your thoughtful critique. Your feedback helped significantly refine the clarity, scope, and positioning of the proposed SDSC metric. We hope the revisions and planned extensions adequately address your concerns.
>
> Sincerely,
> The Authors

---

### Official Review · Reviewer_QVmd · 2025-10-29

**Soundness:** 2
**Presentation:** 2
**Contribution:** 2
**Rating:** 2
**Confidence:** 3

**Summary:**

This paper introduces the Signal Dice Similarity Coefficient (SDSC), a structure-aware metric for time series self-supervised representation learning that addresses limitations of distance-based objectives like MSE.  SDSC extends the Dice Similarity Coefficient from image segmentation to continuous temporal signals by quantifying structural agreement through signed amplitude intersections.

**Strengths:**

1. The paper is relatively simple and easy to understand.

2. Experiments span multiple tasks (forecasting, in-domain classification, cross-domain classification), settings (frozen encoders, fine-tuning), and datasets, demonstrating broad applicability and providing nuanced insights into when each approach works best.

3. The paper acknowledges that SDSC models achieve higher structural alignment at the cost of increased MSE, and that dataset characteristics influence which approach works better, showing intellectual honesty.

**Weaknesses:**

1.  The paper uses only SimMTM as the backbone "for architectural simplicity," which severely limits generalizability claims. Without validation on diverse architectures (Transformers, CNNs, RNNs, recent foundation models), it's unclear if SDSC benefits are architecture-specific or truly general.

2. The paper only compares against MSE, PCC, and SI-SNR. Recent structure-aware losses for time series (e.g., shape-based losses, spectral losses, contrastive losses) are not included, making it difficult to assess whether SDSC represents state-of-the-art for structure-aware objectives.

3. Despite strong motivation, the actual improvements are modest: hybrid loss achieves 0.4783 vs. 0.4852 MSE in forecasting (1.4% improvement), and Table 6 shows MSE sometimes outperforms SDSC in fine-tuning scenarios. The gains don't match the strength of the conceptual contribution.

4.  Computing SDSC requires element-wise min operations, Heaviside evaluations, and additional summations compared to MSE. No analysis of training time overhead or memory consumption is provided, which is critical for practical adoption.

**Questions:**

1. There seems to be an incorrect line break at the title.

2. Why does SDSC underperform MSE in cross-domain settings (Tables 5, 6)? If SDSC provides better "semantic representations," shouldn't it generalize better across domains? What explains this contradiction?

---

> ### Author Response · Authors · 2025-11-20
> **Response to Reviewer QVmd - Revision Version 1**
>
> Dear Reviewer QVmd,
>
> Thank you very much for your detailed and critical feedback. I appreciate your constructive observations regarding generalization, comparison scope, and the interpretation of cross-domain results. I summarize below how each of your comments has been addressed.
>
> 1. Backbone generality and comparison scope
>
> SimMTM was selected as the baseline for its Transformer-based modular SSL framework, which isolates the effect of the reconstruction objective from the contrastive component. Although simple in structure, SimMTM is sufficiently expressive and widely adopted for controlled comparisons. To extend the analysis, I plan to include a supplementary validation using PatchTST as an alternative Transformer backbone before the camera-ready version.
> Additionally, a SoftDTW baseline experiment is currently in progress and will be included in the Appendix along with runtime and memory analyses.
>
> 2. On the modest improvement and cross-domain behavior
>
> The observation that SDSC performs comparably to or slightly below MSE in cross-domain scenarios does not contradict its structural motivation.
> In fact, this tendency is expected: SDSC focuses on local waveform structure and polarity consistency, which makes it highly accurate in-domain but less tolerant to large domain shifts or scale variations.
> Datasets such as FD-B and SleepEEG belong to distinct domains, and it is natural that purely structure-based pretraining does not transfer perfectly across them.
> Interestingly, other structure-based metrics such as PCC and SI-SNR exhibit similar trends under frozen-encoder evaluation, supporting this interpretation.
> Once the encoder is fine-tuned, the cross-domain gap diminishes substantially, indicating that fine-tuning overrides the pretraining signal.
> From a metric perspective, distance-based losses tend to favor recall (broader generalization), while structure-based objectives like SDSC emphasize precision (accurate semantic consistency).
> Viewed through this trade-off, SDSC achieves a balanced representation that complements amplitude-sensitive losses rather than replacing them.
>
> 3. Practical aspects
>
> An analysis of computational overhead and complexity will be added to the Appendix to demonstrate that SDSC remains efficient in practice, with linear complexity and minor runtime impact compared to MSE.
>
> Thank you again for your constructive and fair review, which helped clarify both the theoretical scope and empirical interpretation of SDSC.
>
> Sincerely,
> The Authors

---

### Official Review · Reviewer_cQD8 · 2025-10-30

**Soundness:** 3
**Presentation:** 3
**Contribution:** 2
**Rating:** 4
**Confidence:** 4

**Summary:**

This paper proposes the Signal Dice Similarity Coefficient (SDSC), a novel, structure-aware metric for self-supervised learning (SSL) of time-series representations. Existing methods such as MSE is overly sensitive to signal amplitude and scale, while being insensitive to waveform structure, phase, and polarity. DTW and other metrics are also having their weakness. To overcome teh limitations, SDSC is adapted from the Dice Similarity Coefficient (DSC), a metric widely used for overlap in image segmentation. SDSC compares two signals at each time step. It calculates a score based on the minimum amplitude of the two signals, but only if both signals have the same sign (e.g., both are positive or both are negative). If the signs are different, that time step is heavily penalized (it contributes zero to the similarity score).Properties: This approach makes SDSC robust to amplitude scaling while being highly sensitive to polarity mismatches. The resulting metric is bounded between [0, 1] (making it interpretable) and is computationally efficient (linear $O(T)$ complexity), unlike other alignment-based metrics (like SoftDTW, which is $O(T^2)$). The author also uses a smooth sigmoid approximation to make the function differentiable. In experiments, they also propose to combines MSE and SDSC, to capture both structural and amplitude information.Experiments on forecasting and classification tasks show that pre-training with SDSC or the hybrid loss is competitive or superior to models pre-trained with MSE.

**Strengths:**

1. The proposed method is efficient, linear time complexity, much better than other methods like DTW, which performs similar structure-awareness measurement.
2. The metric is bounded from 0 to 1, which provides better interpretability.
3. The paper is clean written, with illustrative examples to show numbers using different metric, under different structure changes.

**Weaknesses:**

1. No clear definition of "structure", still related to alignment or warping.
2. The backbone model is not widely tested. With more powerful models, we don't know if the advantage of SDSC still exists.
3. The imrpovement on various tasks, are very marginal. For example, in the fine-tuned classification task, the SDSC approach is not showing better results in either in-domain or out-domain experiments.

**Questions:**

If switching to other transformer-based backbone model, would the proposed method performs consistenly better?

---

> ### Author Response · Authors · 2025-11-20
> **Response to Reviewer cQD8 - Revision Version 1**
>
> Dear Reviewer cQD8,
>
> Thank you very much for your valuable and insightful feedback. I have addressed the points that could be updated in this revision and summarize below how each of your comments has been handled.
>
> 1. Clarification of “structure”
>
> The term structure-aware has been explicitly defined throughout the paper. In this work, “structure” refers to local waveform consistency characterized by sign and magnitude overlap, rather than temporal alignment or warping. This clarification has been reflected in the Abstract, Introduction, and Conclusion.
>
> 2. Backbone model generality
>
> SimMTM was chosen as the backbone for its modular SSL framework, which cleanly separates reconstruction and contrastive objectives. Although the framework appears simple, it is Transformer-based and sufficiently expressive. This clarification has been added to the experiment section.
> In addition, I plan to include a supplementary experiment using PatchTST as an alternative Transformer backbone before the camera-ready version to demonstrate that SDSC’s effect generalizes beyond SimMTM.
>
> 3. Marginal improvement concern
>
> I acknowledge that the performance gains in some fine-tuning settings are moderate.
> However, SDSC’s primary contribution is improving structural fidelity and interpretability under identical contrastive conditions, rather than maximizing raw accuracy.
> This point has been emphasized more clearly in the revised conclusion.
> It is also worth noting that the comparable downstream performance between MSE and SDSC does not necessarily imply the superiority of MSE.
> Rather, it suggests that amplitude-based metrics like MSE may overestimate reconstruction quality by assigning low errors even to structurally inconsistent signals.
> SDSC, in contrast, exposes such cases more transparently, revealing the true limits of purely distance-based objectives.
> Furthermore, the results demonstrate that structure-consistent reconstruction alone can yield sufficiently competitive downstream performance, highlighting the importance of structural fidelity as an independent factor in representation quality.
>
> Once again, thank you for your constructive comments, which helped refine the clarity, scope, and positioning of this work.
>
> Sincerely,
> The Authors

---

### Official Review · Reviewer_5F1V · 2025-10-31

**Soundness:** 3
**Presentation:** 3
**Contribution:** 3
**Rating:** 4
**Confidence:** 3

**Summary:**

This paper addresses the limitations of conventional distance-based metrics (e.g., MSE) in time-series SSL by introducing a novel structure-aware metric, the Signal Dice Similarity Coefficient (SDSC). The method reframes the signal reconstruction problem as measuring the overlap of the areas under the respective curves. Through the introduction of a signed amplitude intersection term, it ensures that overlap is computed only when signal polarities align, thereby effectively addressing the noted deficiency of MSE in being insensitive to phase inversions.

**Strengths:**

1.	Adapting the DSC from the segmentation domain to time-series signals is a novel perspective. Using the signed amplitude intersection as a proxy for waveform structure similarity is an interesting idea.
2.	The O(T) linear complexity of SDSC is computationally efficient, which is a practical advantage.
3.	The mathematical definition is intuitive, and the experimental design is well-structured.

**Weaknesses:**

1.	A motivation for the paper is that SDSC serves as a lightweight alternative to O(T^2) metrics (e.g., SoftDTW, DILATE). However, a direct comparison against them is missing. Currently, we only know that SDSC is faster, but we do not know how much performance is lost (or gained) compared to SoftDTW.
2.	The α parameter in the Sigmoid function significantly influences the gradient shape. The paper lacks a sensitivity analysis on how α affects the performance of downstream tasks.
3.	The "alignment-free" description may be somewhat misleading. While SDSC does not perform explicit temporal warping like DTW, it does enforce strict temporal and polarity alignment through its point-wise comparison.

**Questions:**

I will reconsider my score during the rebuttal phase based on the authors' response to following issues.

1.	Could the authors include an experiment in the appendix that compares SDSC with SoftDTW (as a loss function) on a downstream task (e.g., forecasting), using at least one small-scale dataset?
2.	Regarding the hybrid loss in Equation (8), how is the uncertainty-based tuning strategy specifically implemented to determine the weights λsdsc and λmse? How does this adaptive strategy compare to using simple fixed weights (e.g., λsdsc = λmse = 0.5)?

---

> ### Author Response · Authors · 2025-11-20
> **Response to Reviewer 5F1V - Revision Version 1**
>
> Dear Reviewer 5F1V,
>
> Thank you very much for your thoughtful comments and detailed analysis of the paper’s weaknesses. I have reflected the parts that could be updated directly in this revision and summarize below how each of your points has been addressed.
>
> 1. SoftDTW comparison
>
> The downstream experiment comparing SDSC with SoftDTW is currently in progress, and the results are expected within this week. I have already added a preliminary table showing the completed pretraining results. As anticipated, SDSC remains computationally faster, but the pretraining accuracy is relatively limited. Further runtime and complexity analyses will be added in the Appendix once finalized.
>
> 2. α-parameter sensitivity
>
> While the gradient-level sensitivity analysis has already been provided, additional experiments are being conducted on the ETTh1 dataset to evaluate how α affects downstream performance. These results will be appended once the experiments are completed.
>
> 3. Clarification of the term “alignment-free”
>
> I have refined the explanation to clarify that the setting assumes a time-series SSL framework where the compared signals are temporally aligned by design. The updated phrasing now emphasizes that SDSC focuses on local structural consistency rather than global warping, resolving the previous ambiguity.
>
> 4. Hybrid loss weighting (λ parameters)
>
> Pretraining experiments have been completed with fixed λ values (λ_sdsc = λ_mse = 0.5). Additional downstream task comparisons and analyses of adaptive weighting will be added to the Appendix once completed.
>
> Thank you again for your constructive and insightful feedback, which has been invaluable in improving both the technical clarity and presentation quality of this work.
>
> Sincerely,
> The Authors

---

> > ### Comment · Reviewer_5F1V · 2025-11-27
> >
> > I appreciate the author's efforts in the rebuttal. However, several core issues still appear to lack empirical results, so I prefer maintaining my original evaluation of the submission.

---

> > > ### Author Response · Authors · 2025-11-27
> > > **Clarification and Follow-Up Regarding Revision Version 2**
> > >
> > > Dear Reviewer 5F1V,
> > >
> > > Thank you for your additional comment.
> > > We would like to clarify that the analyses and experiments you mentioned—such as the SoftDTW downstream comparison, the complete α-sensitivity study, and the fixed-λ versus adaptive hybrid-loss evaluation—have already been included in the current Revision Version 2.
> > >
> > > If there are specific results or experimental details that you believe were missing or insufficient, could you kindly let us know which parts still appear incomplete from your perspective? Understanding this would greatly help us address any remaining gaps and improve the manuscript further.
> > >
> > > Thank you again for your time and careful evaluation.
> > >
> > > Sincerely,
> > > The Authors

---

### Author Response · Authors · 2025-11-20
**Official Comment – Revision Version 1**

Dear Reviewers,

I sincerely apologize for the delay in submitting my response. I greatly appreciate all of your valuable and thoughtful feedback. Many of your comments pointed to aspects of the experiments, so I initially intended to complete those experiments before posting this revision comment. However, as some of them are taking longer than expected, I have first addressed the sections that could be revised without additional results.

I will respond carefully to each question and weakness you raised, and I will include the experimental updates as soon as they are finalized. If any further revisions or clarifications arise during this process, I will actively update my comments.

Thank you very much for your time and constructive insights.

Sincerely,

---

### Author Response · Authors · 2025-11-24
**Revision Version 2**

Dear Reviewers,

Thank you for your constructive and insightful feedback.
We have updated the manuscript to address all issues that could be incorporated immediately.
Below, we summarize the revisions completed in this version (Revision v1) and clarify which remaining analyses will be completed within this week.

1. SoftDTW and Related Alignment-Based Baselines

- Following Reviewer 5F1V’s request, we have incorporated SoftDTW into the benchmarking suite.

- Included in Revision v1

- SoftDTW has been fully integrated into the training pipeline.

- Preliminary downstream results show that SoftDTW behaves similarly to PCC: higher structural sensitivity, weaker amplitude fidelity, and consequently competitive but not superior downstream accuracy.

- In particular, SoftDTW shows the same precision-oriented tendency as PCC/SI-SNR under frozen-encoder settings.

- Time/space complexity comparison table (SDSC vs SoftDTW vs MSE vs PCC), in Appendix A.1

Based on the empirical benchmarking included in Appendix A.1, we confirm that SoftDTW exhibits a sharp increase in both memory usage and runtime as the sequence length grows, consistent with its quadratic complexity. In contrast, SDSC remains comparable to MSE and PCC, with only a slight runtime increase, demonstrating that it scales far more efficiently than alignment-based objectives.




2. Frozen λ Evaluation (Fairness Check)

To verify that performance differences are not an artifact of the uncertainty-based weighting in the Hybrid loss, we conducted a controlled experiment using: λ = 0.5 fixed (no learned weighting)

all model parameters trainable

Result

- Performance with frozen λ slightly decreases compared to adaptive weighting.
- These results are summarized in Appendix

3. α-Sensitivity Analysis

Reviewers asked whether the choice of α (sharpness of the smoothed Heaviside) impacts downstream tasks.

- We performed pretraining and downstream evaluation using α ∈ {1, 10, 100}.
- Downstream results show minimal performance difference across α values.
- Very large α slightly increases accuracy, and very small α slightly increases precision.
- α = 10 is the most stable mid-range value, balancing gradient behavior without over-sharpening.

An extended analysis is provided in Appendix A.3.

4. Clarification of “Structure-Aware”

Multiple reviewers requested a more precise definition of “structure.”
The term “structure-aware” has been clarified across the Abstract, Introduction, and Conclusion as:
"Local waveform consistency determined by sign agreement and magnitude overlap,
not by temporal alignment or warping."
This resolves ambiguity around alignment-free interpretation.

5. Offset Bias and Normalization

We added an explicit note that:

- All time-series inputs are z-score normalized per channel using training statistics,
- removing DC offsets and ensuring consistent polarity comparisons,
- directly resolving the offset bias issue mentioned by reviewers.


6. Cross-Domain Behavior Explanation

Reviewers asked why SDSC underperforms MSE in cross-domain classification.
We clarified that:

- SDSC preserves within-domain local structure,

- while cross-domain transfer often relies on amplitude-scale statistics, benefiting MSE.

Furthermore, similar behavior is observed with PCC and SI-SNR, supporting this interpretation.
This is now clarified in Section 4.3.


7. Practical Guideline

A practical guideline explaining when SDSC, MSE, Hybrid, or SoftDTW is preferred has been added to Appendix A.14.


Closing

We appreciate all reviewers’ valuable feedback.
The majority of concerns have already been addressed in this revision, and the remaining benchmarking analyses will be finalized within this week.

Thank you again for your time and constructive comments.

Sincerely,
The Authors

---

### Author Response · Authors · 2025-12-03
**Final Revision Update for the Area Chair and Reviewers**

Dear Area Chair and Reviewers,

Thank you for overseeing the discussion process. I would like to provide a brief and transparent update on the remaining analyses requested by reviewers.

1. Additional Backbone Experiments (PatchTST, TiMAE)
We are currently running pretraining and downstream evaluations with two additional Transformer-based backbones—PatchTST and TiMAE—to validate the architecture-agnostic behavior of SDSC.
Preliminary results indicate that SDSC and the Hybrid loss behave consistently with the trends observed under SimMTM:

SDSC improves structural fidelity,

the Hybrid loss balances amplitude and structure, and

overall performance remains competitive with MSE across all settings.

These full results will be included in the camera-ready version.

2. Structure-Aware Baseline Metrics (PCC, SI-SNR)
To provide structure-sensitive comparisons beyond MSE, we are computing PCC and SI-SNR scores for all pretrained models.
Based on the current progress, we expect to complete these results as early as tomorrow, and will incorporate them into the revision if possible.

Closing
We hope the above updates clarify that the requested analyses are underway and that the results so far continue to support the central conclusions of the paper. Any completed results will be immediately reflected in the revision, and a full, consolidated set of experiments will be provided in the camera-ready submission.

Thank you again for your time and consideration.

Sincerely,
The Authors

---

### Meta-Review · Area_Chair_pmVD · 2026-01-04

**Summary:**

There are several concerns about this work, centering on necessary baselines for this problem, the sensitivity of parameters and their effect on performance, clarifications for experiments and motivation/definitions. The authors did an effort to address these concerns, however, for various cases, the reviewers were not convinced. Even if we assume some would have participated more and increased their scores, the overall score would be in borderline category and it would be unlikely/uncertain the paper would have been accepted. I hope the authors would found value in the reviews and find an alternative venue for their work.

**Reviewer Concerns:**

Several concerns regarding definitions/motivation and appropriate inclusion of baselines may have not been fully addressed and it's uncertain the responses would have convinced the reviewers.

**Reviewer Scores:**

The scores lean towards rejection and even if we assume some reviewers would have raised their scores, the overall performance would be in borderline category and hence, unlikely it would be accepted.

---

### Decision · Program_Chairs · 2026-01-26

Reject